# Poisoning among children in Malaysia: A 10-years retrospective study

**Iqdam Abdulmaged Alwan[1]\*, Ali Saeed Brhaish[1], Ammar Ihsan Awadh[2], Asdariah Misnan[3], Nur Arzuar Abdul Rahim[2], Balamurugan Tangiisuran[4], Mohamed Isa Abdul Majid[3]**

**1** Integrative Medicine Cluster, Advanced Medical and Dental Institute, Universiti Sains Malaysia, Kepala Batas, Penang, Malaysia, **2** Department of Pharmacy, Al-Kunooze University College, Basra, Iraq, **3** National Poison Centre, Universiti Sains Malaysia, Gelugor, Penang, Malaysia, **4** School of Pharmaceutical Sciences, Universiti Sains Malaysia, Gelugor, Penang, Malaysia

\* iqdamalwan@gmail.com

## Abstract

### Background

Poisoning commonly occurs among children due to their curiosity, where they tend to explore and investigate their surroundings. They frequently put what they find into their mouths as they do not understand the danger and probably cannot read the warning label. As this issue has not been extensively studied in Malaysia; hence, a retrospective analysis of records was carried out to determine the profile of phone call enquiries regarding poisoning among children at the National Poison Centre (NPC).

### Methodology

The records of all cases of poisoning among children below the age of 18 years were retrospectively reviewed over a period of 10 years from 2006 to 2015. The data on the cases were analysed according to age group and gender, the circumstances and the toxic agent implicated in the poisoning.

### Results

During the 10-year study period, 13,583 calls that met the criteria for this study were referred to the NPC. Of these calls, 62.2% involved children between the age of 0 to 5 years, 9% were children aged between 6 to 12 years, and 28.8% were children between 13 to 18 years. Unintentional poisoning accounted for 96.7% of the incidents involving children between the age of 0 to 5 years, although among the children who were between the age of 13 to 18 years, 76% of the cases were intentional. In all the cases involving children, pharmaceutical agents were the most frequent source of the poisoning. More than 95% of the cases were exposed to poisoning through the oral route.

### Conclusion

Poisoning in children between the age of 0 to 5 years was mainly unintentional, while poisoning in children between the age of 13 to 18 years was mainly intentional, where

**Data Availability Statement:** The data of the current study are fully owned by the National Poison Centre of Malaysia. Further information is available from, URL: https://prn.usm.my/. Noor Afiza Md. Rani, Research Officer, Email:

noor_afiza@usm.my. The authors did not have any
special access or privileges that others would not
have.

**Funding:** The author(s) received no specific
funding for this work.

**Competing interests:** The authors have declared
that no competing interests exist

pharmaceutical and household agents were responsible for more than two-thirds of the poi-
soning cases. Most of these incidents could have been prevented if protective measures,
such as child-resistant enclosures, had been implemented and if the parents and guardians
had been educated about preventive measures, such as keeping poisoning agents out of
the reach of children.

## Introduction

Poisoning among children is a significant public health issue around the world. It is reported
to be the fourth leading cause of accident-related mortalities [1]. It is also thought to be one of
the main causes of morbidity in both developing and developed countries [2]. Based on the
World Health Organization (WHO) statistics, acute poisoning is responsible for more than
45,000 deaths every year among children and youths below the age of 20 years" [3]. The under-
lying causes of poisoning differ across countries, depending, amongst others, on the local cus-
toms and beliefs, demography, socio-economic status of the population in that area, and level
of education. Besides, poisoning patterns may vary with respect to the age and gender of the
individuals [4].

In the United States, the highest incidence of exposure to poisoning occurs among 1- and
2-year-old children. Unintentional poisoning is common among those aged 12 years and
below, and the incidence of intentional poisoning is remarkably high among those aged 12
years and above [5]. In Asian countries, for instance, India, Sri Lanka and Taiwan, it is believed
that there is poor and inadequate surveillance of data on poisoning in comparison to the devel-
oped countries. About 90% of the poisoning cases are selectively reported by physicians to the
poison centres rather than by the lay public [6]. A study in Thailand stated that the majority of
poisoning incidents occur among children aged 5 years and below. Although exposure to poi-
soning among new-born babies up to twelve-year olds is unintentional, the vast majority of
exposures in the older groups are intentional [7, 8].

The National Poison Centre of Malaysia was founded in 1995 by the Drug and Poison
Information Service (DPIS), which remains the only poison information centre in the country.
DPIS is the main source of information on poisoning risks and toxicity. It serves healthcare
professionals and the general populace with the relevant information when they inquire about
such matters. The NPC operates 24/7, whereby pharmacists, trained in toxicology and special-
ize in providing drug and poison information services, offer advice to enquirers, usually via
telephone calls. It also has experienced consultants on hand to deal with complicated cases.
The centre handles about 4000–6000 enquiries on an annual basis. Information on paediatric
poisoning in Malaysia is limited. Thus, given the current statistics, this study will attempt to
present a clear picture to public authorities so as to spark actions for solutions and for future
researchers to conduct further research into poisoning among children in Malaysia. Hence,
the objective of this study is to determine the pattern, highlight the main causes and types of
poisoning, and describe the sociodemographic characteristics of poisoning among children in
Malaysia.

## Methodology

The National Poison Centre (NPC) uses a standard and robust recording system for the proper
documentation of information with regard to all incoming calls and enquiries, such as the
enquirer's name and address, product / poison information (chemical name), patient's age,

sex, route and duration of exposure, mode and symptoms of poisoning, treatment already provided and necessary queries about the patient. All calls to the NPC are recorded into a voice logging and recording system and the recorded audio is retrievable for documentation, training and auditing purposes.

This study is a retrospective review of poisoning cases among children referred for enquiries to the NPC from 2006 to 2015. Records were extracted for children aged 18 years and below in accordance with the definition of child given by the United Nations Educational, Scientific and Cultural Organization as any person below the 18 years of age [9]. The children were divided into three age groups: 0 to 5 years, 6 to 12 years, and 13 to 18 years. This classification of age is consistent with the reports of the American Association of Poison Control Centers' (NPDS) [5, 10]. The calls were manually reviewed for inclusion. A single poisoning event sometimes prompted several calls to the NPC (e.g., a call from a member of the public, then from a triage nurse, then from a doctor). Such subsequent calls were excluded for the purposes of this study, with the exception of caller background counts and symptoms analyses, where re-calls were included (as calls from hospitals provided more details about symptoms and dispositions).

The Ethical approval with the registration number: (NMRR-15-273-13892) was obtained from the National Institutes of Health and Medical Research Ethics Committee, Ministry of Health Malaysia. In order to protect the privacy of the patients, the personal data were erased before the descriptive analysis was carried out. The descriptive analysis for this study was performed by using the Statistical Package for Social Sciences (SPSS) version 22.0 to describe the results in terms of frequencies and percentages.

## Results

During the 10-year period (2006–2015), the NPC received 13,583 calls concerning poisoning cases among children aged 18 years and below.

The number of calls constantly increased from 493 cases in 2006 to a peak of 2,098 cases in 2011, before dropping to 1,270 cases in 2012. Then again, a similar pattern of increase occurred, where the number of cases increased from 1,662 in 2013 to 1,841 in 2015, as shown in Fig 1. It should be noted that the slope in 2012 was merely because the NPC reduced its operating hours from 24 hours to 8 hours a day.

Based on gender, males outnumbered females, where 5,869 (43.2%) cases involved males and 5,440 (40.1%) involved females, while 2,274 (16.7) cases were of unknown gender. The vast majority of the incidents occurred among children aged 0 to 5 years (62.2%), followed by those aged 13 to 18 years (28.8%), and 6 to 12 years (9%). A male predominance was evident among those between 0 to 5 years of age (47%), while a female predominance was common in those aged between 13 and 18 years (62.6%). The type of incident varied according to age group, where the majority of the cases involving those aged under 12 years were unintentional poisoning, where the (0–5) and (6–12) age groups formed 96.7% and 82.5% of the cases, respectively. On the other hand, intentional exposure was highly prevalent among those aged between 13 to 18 years (76%) (see Table 1). Adverse reactions accounted for about 0.7% among those aged between 0 to 5 years, 2.2% among those aged between 6 to 12 years, and 0.7% among those aged between 13 to 18 years. The majority of the cases involving exposure to poisoning occurred at home (96%) and the highest incidence was by way of the oral route, which accounted for more than 96% of the cases, followed by bites/stings (1.5%), and inhalation (1.1%), as shown in Table 1.

From the data collected for the 10-year period, it was found that the top three categories of poisoning sources were pharmaceutical agents, household agents and pesticides.

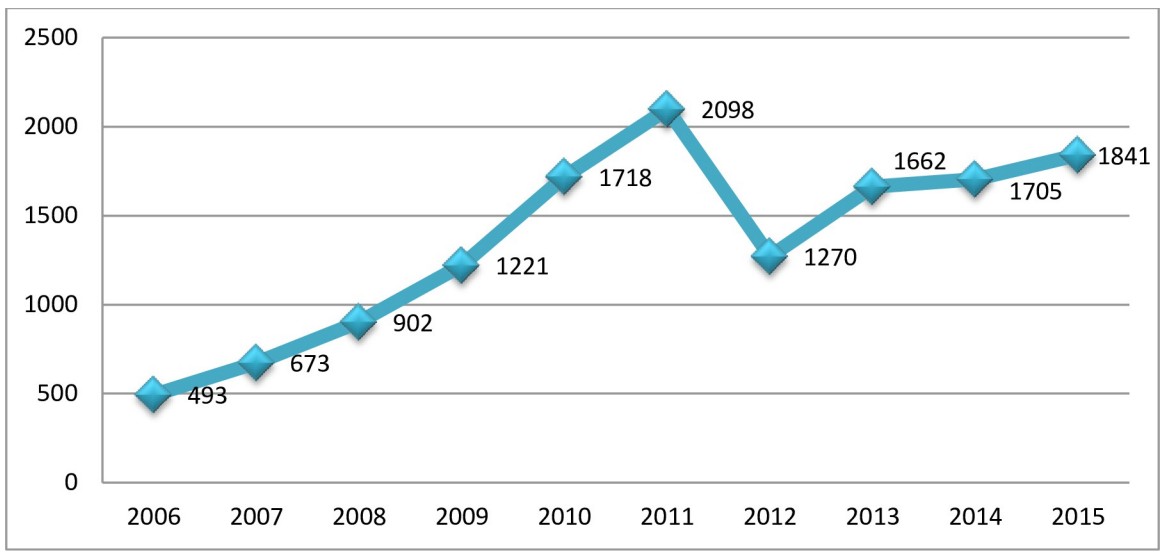

**Fig 1. Total number of calls received by the National Poison Centre from 2006–2015 involving individuals 18 years and under.**

Table 1. The characteristics of poisoning according to the age groups of the children.

| Age group (years) | 0–5 | 6–12 | 13–18 | Total |
|---|---|---|---|---|
| | n (%) | n (%) | n (%) | n (%) |
| The number of cases | 8454 (62.2) | 1211 (9) | 3918 (28.8) | 13583 (100) |
| Gender | | | | |
| Male | 3,976 (47.0) | 655 (54) | 1,238 (31.6) | 5,869 (43.2) |
| Female | 2,622 (31.0) | 364 (30.1) | 2,454 (62.6) | 5,440 (40.1) |
| Unknown | 1,856 (22.0) | 192 (15.9) | 226 (5.8) | 2,274 (16.7) |
| The manner of poisoning | | | | |
| Un-Intentional | 8,177 (96.7) | 999 (82.5) | 914 (23.3) | 10,090 (74.3) |
| Intentional | 215 (2.5) | 185 (15.3) | 2,978 (76) | 3,378 (24.9) |
| Adverse Reaction | 62 (0.7) | 27 (2.2) | 26 (0.7) | 115 (0.8) |
| Route of exposure | | | | |
| Ingestion/oral | 8,203 (97) | 1,082 (89.3) | 3,739 (95.4) | 13,024 (95.9) |
| Bite/sting | 70 (0.8) | 67 (5.5) | 73 (1.9) | 210 (1.5) |
| Inhalation | 53 (0.6) | 25 (2) | 71 (1.8) | 71 (1.8) |
| Other[1] | 128 (1.5) | 37 (3) | 35 (0.9) | 200 (1.5) |
| Place of exposure | | | | |
| Home | 8,267 (97.8) | 1,132 (93.5) | 3,672 (93.7) | 13,071 (96.2) |
| Hospital | 80 (0.9) | 22 (1.8) | 19 (0.5) | 121 (0.9) |
| Workplace | 10 (0.1) | 8 (0.7) | 39 (1) | 57 (0.4) |
| Other[2] | 97 (1.1) | 49 (4) | 188 (4.8) | 334 (2.5) |

[1]Include Injection 66(0.5) Cutaneous 64(0.5) Ocular 14(0.1) Mucosal 14(0.1) Placental 7(0.05) Otic 1(0.007) Unrecorded 34(0.25).

[2]Include Open Place 57(0.4) Academic Institutions 48(0.4) Nursing homes 7(0.05) Enclosed Public Place7 (0.05) Institutions (prison, military) 8(0.06) Unknown 207(1.5).

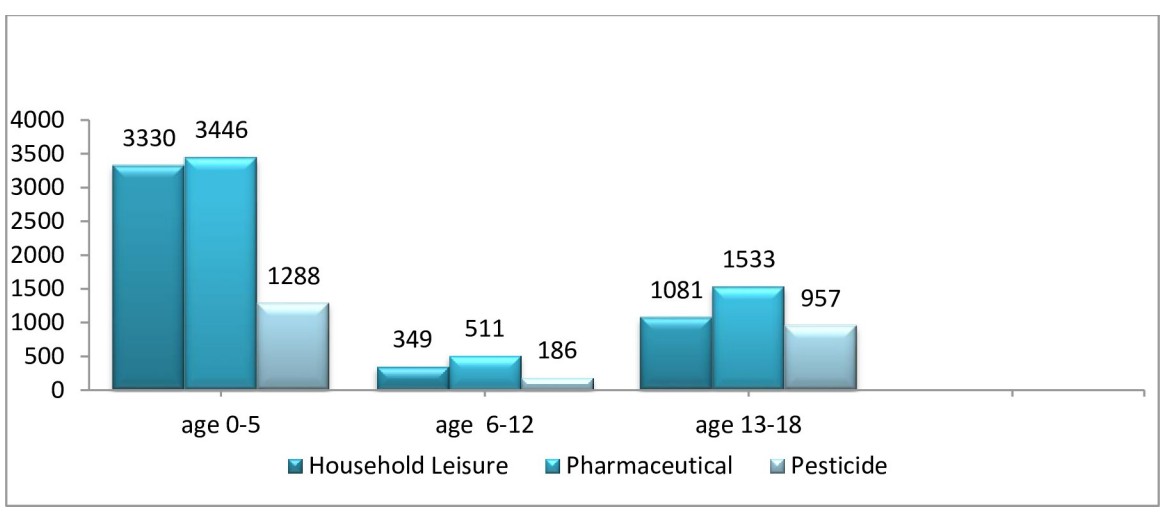

**Fig 2. Distribution of number of poisoning cases in age group according to type of poisoning.**

Pharmaceutical agents were the most frequent in all the age groups. The most frequent non-pharmaceutical agents in all the age groups were household agents, as illustrated in Fig 2. Exposure to therapeutic agents and household products together accounted for 10,250 cases or 75% of all reported exposures.

When the most frequent poisoning agents within the aforementioned categories in the Fig 2 were identified according to age groups, it was found that within the pharmaceutical group, topical agents, including aromatherapy oils and analgesics, were significant across all ages. Among the household agents, cleaning products were also seen to be significant, while insecticides and herbicides were the most common pesticides across all ages (Table 2).

The top 10 agents that were commonly the cause of child poisoning were cleansing agents (2,087 cases or 15.4%), followed by topical agents (1,511 cases or 11.1%), while the least

**Table 2. The most common poisoning agents in children according to age group.**

| Age group (years) | 0–5 | 6–12 | 13–18 | Total |
|---|---|---|---|---|
| **Agent** | n (%) | n (%) | n (%) | n (%) |
| **Pharmaceuticals:** | **3446 (62.8)** | **511 (9.3)** | **1533 (27.9)** | **5490 (100)** |
| **Topical agent** | 1,137 (75.2) | 143 (9.5) | 231 (15.3) | 1,511 (27.4) |
| Analgesic | 309 (42.7) | 54 (7.5) | 360 (49.8) | 723 (13.1) |
| Psychiatric | 309 (53.2) | 69 (11.9) | 203 (34.9) | 581 (10.5) |
| Other pharmaceuticals | 1691 (62.7) | 266 (9.9) | 739 (27.4) | 2696 (49) |
| **Household agents:** | **3330 (70)** | **349 (7.3)** | **1081 (22.7)** | **4760 (100)** |
| Cleaners | 1,186 (56.8) | 137 (6.6) | 764 (36.6) | 2,087 (43.8) |
| Solvents | 536 (79.2) | 61 (9.0) | 80 (11.8) | 677 (14.2) |
| Cosmetic/Personal care | 363 (77.7) | 25 (5.4) | 79 (16.9) | 467 (9.8) |
| Other households | 1328 (86.9) | 100 (6.5) | 101 (6.6) | 1529 (32.2) |
| **Pesticides:** | **1288 (53)** | **186 (7.6)** | **957 (39.4)** | **2431 (100)** |
| Insecticide | 332 (39.6) | 89 (10.6) | 418 (49.8) | 839 (34.5) |
| House Insecticide | 502 (85.2) | 22 (3.7) | 65 (11) | 589 (24.2) |
| Herbicide | 120 (22) | 44 (8) | 382 (70) | 546 (22.5) |
| Other pesticides | 334 (73) | 31 (6.8) | 92 (20.2) | 457 (18.8) |

**Table 3. Top 10 poisoning agents list.**

|  | Agent | Total n (8929) | Percentage (%) (65.7) |
|---|---|---|---|
| **1** | Cleaner | 2,087 | (15.4) |
| **2** | Topical agent | 1,511 | (11.1) |
| **3** | Other household | 909 | (6.7) |
| **4** | Insecticide | 839 | (6.2) |
| **5** | Analgesic | 723 | (5.3) |
| **6** | Solvents | 677 | (5.0) |
| **7** | House Insecticide | 589 | (4.3) |
| **8** | Psychiatric medications | 581 | (4.3) |
| **9** | Herbicide | 546 | (4.0) |
| **10** | Cosmetic/Personal care | 467 | (3.4) |

common were cosmetic/personal care products (467 cases or 3.4%). The top 10 items alone constituted 66% of all exposures (Table 3).

## Discussion

The current study discussed one of the most challenging medical issues, which is lacking in sufficient epidemiological elucidation in Malaysia. It involved an investigation into more than ten thousand medical records of poisoning incidents among different age groups (0 to 18 years).

Based on the calls that were made to the NPC, the number of poisoning incidents among children was on the rise. However, there was a remarkable decline in 2012, and this was only because the NPC had temporarily reduced its operating hours from 24/7 to regular office hours for six months.

The current study showed that poisoning among children is more common in boys. A similar pattern was observed in many studies worldwide, where boys outnumber girls when it comes to exposure to poisons [4, 10, 11]. One of the possible reasons for this could be because boys are highly active compared to girls. However, the exact reason is unknown [4].

Mowry et al. reported that about half of all exposures to poisons in the United States occur among children who are below the age of 6 years [10]. Bhat et al. stated that 60.68% of the cases of poisoning in India occur among children below the age of 6 years [12]. Al-Barraq and Farahat, also stated that the majority of the poisoning cases among children in Saudi Arabia occur among those below the age of 5 years [13]. Oliveira and Suchara stated that the highest percentage of poisoning in Brazil is among children below the age of 4 years [14]. The results of all these studies were consistent with the findings of this research, which reported that 62,2% of the poisoning cases among children in Malaysia occurred among those between the age of 0 to 5 years.

Various studies have reported that most of the cases of poisoning among children below the age of 6 years are accidental. In contrast, most of the cases among adolescents are intentional [15, 16]. This fact was reaffirmed by the results of this study, where accidental poisoning was more frequent among children between the age of 0 to 5 years, whereas intentional poisoning was predominant among children above the age of 13 years, which might be attributed to the fact that children below the age of 5 years are inquisitive and have a high tendency to explore their surroundings [17]. The current study revealed that the top three causes of poisoning were pharmaceutical agents, followed by household products, and thirdly, pesticides, and this was consistent with other studies where medications and household products were the predominant causes [2, 13, 18–20]. The reasons for this are the availability and accessibility of

these products due to the lack of safe storage measures [21]. It was found that in Malaysia, the most common drugs associated with poisoning in children were topical agents, followed by analgesics. This was in contrast to the findings of other studies abroad, which stated that analgesics are the most common drugs related to poisoning among children [18, 22, 23]. The reason for this inconsistency is that topical agents, including massage oils like *minyak kayu putih* and *minyak cap kapak* are commonly used in Malaysia and are available in every home, where they are often placed on low shelves that can be easily reached by children. In addition, topical and analgesic agents are easily accessible and available as over-the-counter preparations. In Malaysia, medicines are mainly fall under two classes, controlled medicines and over the counter medicines. Over the counter drugs are ubiquitous and they can be easily obtained from any shop or supermarket [24]. In addition, there are no regulations on the quantity of the product that can be taken by an individual at one time. The drugs and other poisonous substances are regulated under the 'Poison Act', which consists of four groups: Group A poisons contains highly toxic substances that are prohibited to sell, group B prescription drugs, Group C Behind the Counter drugs and group D poisons are chemicals used in the laboratory. Non-scheduled drugs exempted from Poison Act 1952 are considered as non-scheduled poisons [25]. Further restrictions on the quantity are required to reduce the incidence of poisoning caused by OTC medications.

Among household products, it was found that cleaners were the most common agents of poisoning. This finding was similar with that of previous studies in the United States and Qatar, where cleansing agents in the household products group are common sources of poisoning [10, 23], the reason being that these products are usually stored in cupboards, which can be easily accessed by children.

According to a systematic review in Thailand, insecticides are the most common agents of poisoning within the pesticides group [26]. The current study made a similar finding.

Özdemir et al. stated that in Turkey the most exposure to poisoning among children takes place in the home and more than 90% of such cases are by the oral route [15]. Lin, Liu et al. from Taiwan showed that all children are exposed by the oral route and the most common location for poisoning is the home [27]. This is consistent with the results of the current study, where most of the incidents of poisoning occurred in the home, and 95% of the poisoning cases were by the oral route, the reason being that children spend most of their time inside the home. Also, the implicated agents in this study, such as medications and household products, are available inside the homes for domestic use.

It is also believed that this prevalence of poisoning among children could have been prevented if certain measures had been implemented. These measures involve educating the parents, as it is crucial to raise awareness among parents of the need to maintain a safer environment for their children. Moreover, packaging that is designed for safety could be helpful as well. For instance, the use of child-resistant caps for all poisonous substances may ensure their safety and reduce the possibility of exposure.

This was a retrospective study of the records of the NPC and, therefore, it was subject to several additional limitations that should be noted explicitly. For example, not all cases of poisoning among children in Malaysia are reported to the NPC and this may give rise to a systematic error. In addition, there was unclassified gender category marked as (unknown), this was due to incomplete information about the gender in the original NPC record form. This study also lacked information on the clinical outcomes of the reported cases, and it was unclear as to the severity of the poisoning, and whether all the reported cases were manageable or not, or were fatal or non-fatal.

Despite the abovementioned limitations, this study was based on results gathered over a period of 10 years and were derived from a relatively large base sample. These results

emphasize the importance of developing a national surveillance strategy for monitoring and managing poisoning events in Malaysia.

## Conclusions

It can be concluded from the study that accidental poisoning is more frequent among children between the ages of 0 to 5 years. Over-the-counter medications, such as topical and analgesic drugs, and household cleaning agents were the most frequent agents of toxicity among children. Additional population-based studies should be conducted to increase the understanding of the epidemiology of childhood poisoning in Malaysia. Appropriate strategies and legislation should be implemented to regulate the sale of over-the-counter medications and ensure that potentially dangerous chemicals are sold in child-resistant containers.

## Acknowledgments

The authors would like to express their appreciation to their colleagues and staff in the National Poison Centre Malaysia for their contribution to this work, with special thanks to statistician from the National Poison Centre, Noor Afiza Abdul Rani for her contribution in the statistical analysis.

## Author Contributions

**Conceptualization:** Iqdam Abdulmaged Alwan, Ali Saeed Brhaish, Ammar Ihsan Awadh, Asdariah Misnan, Balamurugan Tangiisuran, Mohamed Isa Abdul Majid.

**Data curation:** Iqdam Abdulmaged Alwan, Ali Saeed Brhaish, Asdariah Misnan, Balamurugan Tangiisuran, Mohamed Isa Abdul Majid.

**Formal analysis:** Iqdam Abdulmaged Alwan, Ali Saeed Brhaish, Asdariah Misnan, Balamurugan Tangiisuran.

**Funding acquisition:** Iqdam Abdulmaged Alwan, Ammar Ihsan Awadh.

**Investigation:** Iqdam Abdulmaged Alwan, Ali Saeed Brhaish, Ammar Ihsan Awadh, Asdariah Misnan, Nur Arzuar Abdul Rahim, Balamurugan Tangiisuran.

**Methodology:** Iqdam Abdulmaged Alwan, Ali Saeed Brhaish, Ammar Ihsan Awadh, Asdariah Misnan, Balamurugan Tangiisuran, Mohamed Isa Abdul Majid.

**Project administration:** Iqdam Abdulmaged Alwan, Ali Saeed Brhaish, Asdariah Misnan, Balamurugan Tangiisuran, Mohamed Isa Abdul Majid.

**Resources:** Iqdam Abdulmaged Alwan, Ali Saeed Brhaish, Ammar Ihsan Awadh.

**Software:** Iqdam Abdulmaged Alwan, Ali Saeed Brhaish.

**Supervision:** Ammar Ihsan Awadh, Asdariah Misnan, Nur Arzuar Abdul Rahim, Balamurugan Tangiisuran, Mohamed Isa Abdul Majid.

**Validation:** Iqdam Abdulmaged Alwan, Ali Saeed Brhaish, Ammar Ihsan Awadh, Balamurugan Tangiisuran.

**Visualization:** Iqdam Abdulmaged Alwan, Ali Saeed Brhaish, Ammar Ihsan Awadh, Nur Arzuar Abdul Rahim.

**Writing – original draft:** Iqdam Abdulmaged Alwan, Ali Saeed Brhaish, Nur Arzuar Abdul Rahim.

**Writing – review & editing:** Iqdam Abdulmaged Alwan, Ammar Ihsan Awadh.

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
