## [Decision Letter · Decision Letter 0]

19 Jan 2022

PONE-D-21-32551Poisoning Among Children in Malaysia: A 10-Year ReviewPLOS ONE

Dear Dr. Iqdam Alwan

Thank you for submitting your manuscript to PLOS ONE. After careful consideration, we feel that it has merit but does not fully meet PLOS ONE’s publication criteria as it currently stands. Therefore, we invite you to submit a revised version of the manuscript that addresses the points raised during the review process.

Please submit your revised manuscript by  Mar 05 2022 11:59PM If you will need more time than this to complete your revisions, please reply to this message or contact the journal office at plosone@plos.org. Please include the following items when submitting your revised manuscript:A rebuttal letter that responds to each point raised by the academic editor and reviewer(s). You should upload this letter as a separate file labeled 'Response to Reviewers'.A marked-up copy of your manuscript that highlights changes made to the original version. You should upload this as a separate file labeled 'Revised Manuscript with Track Changes'.An unmarked version of your revised paper without tracked changes. You should upload this as a separate file labeled 'Manuscript'.

We look forward to receiving your revised manuscript.

Kind regards,

Shailja Sharma, MD Biochemistry

Academic Editor

PLOS ONE

Journal Requirements:

Reviewers' comments:

Reviewer #1:

Title of the study is “Poisoning Among Children in Malaysia: A 10-Year Review”. Out of the five key words three are a repeat from the title. It is suggested to authors to replace these three repeated keywords, to increase the searchability of the article over internet, if published. The keywords should preferably from the MeSH Vocabulary database.The introduction part should be combined in form of two or three paragraphs. The vancouver style should be used for references.

The manuscript requires the following grammatical corrections:

Line 37-38:

“Poisoning commonly occurs among children as a consequence of their curiosity, where they have a tendency to explore and investigate their surroundings.” Should be replaced by “Poisoning commonly occurs among children due to their curiosity, where they tend to explore and investigate their surroundings.”

Line 76-78:

“Based on statistics by the World Health Organization (WHO), acute poisoning is responsible for more than 45,000 deaths every year among children and youths below the age of 20 years” should be replaced by “Based on the World Health Organization (WHO) statistics, acute poisoning is responsible for more than 45,000 deaths every year among children and youths below the age of 20 years.”

Line 91:

Remove comma after Although

Line 96-100:

Replace “DPIS is the main source of information on poisoning risks and toxicity, and it serves healthcare professionals and the general populace with the relevant information when they enquire about such matters. The NPC operates 24/7, whereby pharmacists, who are trained in toxicology and who specialize in providing drug and poison information services, offer advice to enquirers, usually via telephone calls.”

With

“DPIS is the main source of information on poisoning risks and toxicity. It serves healthcare professionals and the general populace with the relevant information when they inquire about such matters. The NPC operates 24/7, whereby pharmacists, trained in toxicology and specialize in providing drug and poison information services, offer advice to enquirers, usually via telephone calls.”

Line 131:

Replace “Ethic” with “Ethics”

Line 144:

Replace “increased constantly” with “constantly increased”

Line 252:

Remove “in nature”

Line 256:

‘This’ is an unclear antecedent. Please rectify.

Line 323-325:

Replace “Appropriate strategies and legislations should be implemented to regulate the sale of over-the-counter medications and to ensure that potentially dangerous chemicals are sold in child-resistant containers.” with “Appropriate strategies and legislation should be implemented to regulate the sale of over-the-counter medications and ensure that potentially dangerous chemicals are sold in child-resistant containers.”

Reviewer #2: This study aimed to retrospectively analyze the demographic data, types and manner of acute poisoning among children based on telephonic inquiries at NPC. The sample size for studying the profile of acute poisoning in children is adequate. However, the manuscript requires clarifications and a few revisions, as mentioned below.· Title can be changed to Poisoning Among Children in Malaysia: A 10-years retrospective study.· The standard definition of 'child' given by WHO or UNESCO should be included. In the abstract, you have mentioned the inclusion of samples of age 19 years and below, whereas, in the methodology of the manuscript, you have mentioned the samples of age 18 years and below. Please clarify the disparity.· Also, you have subdivided the age groups as below 6 years, 6 to 12 years and above 12 years. Table 1 has categories of 0 to 5 years, 6 to 12 years and 13 to 18 years. Please maintain uniformity. Also, clarify the reason to subdivide the age group into these three categories.· Table 1 can be subdivided into more tables as extensive information has been clustered, which is confusing.· In table number 1, 'character' can be replaced by ‘the number of cases’; 'Incident type' can be replaced by 'the manner of poisoning'.

· In table number 1, the other category in route of exposure should be subdivided in the table only.· Please specify the reasons for gender remaining unknown in about 2000 cases.· Legal scenarios in Malaysia pertaining to various poisoning agents and therapeutic agents can be discussed. The statistical analysis should have been performed appropriately and rigorously. 

---

## [Author Response · Author response to Decision Letter 0]

3 Mar 2022

Dear editors and reviewers,

Thank you for getting back to us, we have responded to all the comments raised by the academic reviewers and changes have been made accordingly. We have submitted a track changes copy of the revised manuscript addressing each comment individually, a revised manuscript file and a letter with responds to the reviewers comments. We would also like to express our acknowledgement and appreciation to the editors and reviewer for their efforts in improving the quality of our manuscript.

Yours faithfully,

Iqdam Alwan

---

## [Editor Report · Decision Letter 1]

28 Mar 2022

Poisoning Among Children in Malaysia: A 10-years retrospective study

PONE-D-21-32551R1

Dear Dr. Iqdam Abdulmaged Alwan

We’re pleased to inform you that your manuscript has been judged scientifically suitable for publication and will be formally accepted for publication once it meets all outstanding technical requirements.

Kind regards,

Shailja Sharma, MD Biochemistry

Academic Editor

PLOS ONE
---

## [Editor Report · Acceptance letter]

20 Apr 2022

PONE-D-21-32551R1 

Poisoning Among Children in Malaysia: A 10-years retrospective study 

Dear Dr. Alwan:

I'm pleased to inform you that your manuscript has been deemed suitable for publication in PLOS ONE. Congratulations! Your manuscript is now with our production department. 

Kind regards, 

on behalf of

Dr. Shailja Sharma 

Academic Editor

PLOS ONE